# Towards a Rubric for Stimulating and Evaluating Sustainable Learning

**Judith Gulikers** * and **Carla Oonk**

Education and Learning Sciences, Wageningen University and Research, P.O. Box 8130, 6700 EW Wageningen, The Netherlands; Carla.oonk@wur.nl
* Correspondence: Judith.gulikers@wur.nl; Tel.: +31-(0)-317-484332

**Abstract:** Preparing students for dealing with sustainability issues is a challenge in the field of education. This is a challenge because we don't know exactly what we are educating for, as there are no defined answers or outcomes to the issues; the future is unpredictable. Dealing with these issues requires crossing boundaries between people coming from different 'practices', e.g., disciplines, cultures, academia versus society, thereby making the learning and working process a challenging but critical learning experience in itself. We argue that education for sustainability should not primarily focus on student content knowledge or development of certain products or answers. It should focus on stimulating students to go through boundary-crossing learning *processes* critical for getting a grip on the unpredictable future. This allows students to learn to work with 'others' around the boundaries, and thereby to develop the ability to co-create new knowledge and work towards innovation or transformation for sustainable practice. Building on the boundary crossing theory and using mixed methods and interventions, this design-based study iteratively develops a boundary crossing rubric as an instrument to operationalise student learning in transdisciplinary projects into concrete student behaviour. This rubric in turn can explicate, stimulate and assess student learning and development in transdisciplinary sustainability projects.

**Keywords:** sustainable learning; boundary crossing; sustainable assessment; stakeholders; student learning; wicked problems

## 1. Introduction

Sustainability problems are inherently complex, ill-defined, contested and lack definite solutions. We call these problems 'wicked' problems [1], stressing that both the problem and its solution(s) are not clear and keep changing whenever we try to define them, like global warming, food security or biodiversity loss. Wals [1] (p. 17) demarcates wicked problems from simple and complex problems. According to Wals, wicked problems are characterised by their resistance to definition, having no right or wrong answers, and their unfamiliar, ambiguous, chaotic nature, in which conflicts of interests among multiple stakeholders are inevitable [2]. Dealing with these wicked issues requires collaboration and meaning-making between these different stakeholders [3]. Various stakeholders represent different 'practices', e.g., different academic disciplines, different cultures and/or different institutional contexts. Sustainability education should prepare students to deal with these wicked problems and their diverse involved stakeholders in their future [2,4]. Lönngren and Svanstöm [5] say that for students to contribute to these problems, they need to develop the ability to address and understand sustainability problems in a holistic and integrative manner, while considering the normative context of sustainable development. This requires, for example, the ability to reconcile conflicting goals, multiple forms of problem representation and solution methods [6], and even more complex sustainability competencies like systems thinking and normative competencies [7,8]. Ceulemans et al. [9] showed that explicit

integration of sustainable development into higher education curricula, at least in Flanders, is still scarce. Svanström et al. [10] showed in their analysis of various educational programs for sustainable development that the learning objectives varied in number and kind, but that "the integration of different perspectives" was one commonality found in all programs. Filho and colleagues [4] moreover stress that higher education should encourage more multi-stakeholder dialogue between students and (non-academic) organisations, confronting students with various aspects (e.g., economic, social, cultural or environmental) relevant for sustainable development.

This requires transdisciplinary education [11] in which students work on real world wicked problems, as well as collaborate across disciplinary and institutional boundaries to work towards knowledge co-creation [12], innovation and transformation [1]. Though transdisciplinary education is sparsely emerging in higher education [3,11], most educational practices address well-structured problems that do not require multi-stakeholder interaction, probably because developing, implementing and assessing education that fosters sustainability competencies is a challenge for teachers and students [5,7]. Teachers are not educated in an inter- or transdisciplinary manner, or might not feel able to integrate sustainable development issues into their education [4,9]. Various reviews on assessment tools have also aimed to compare, evaluate or audit sustainability programs [13,14], and have shown that the actual design and activities in the areas of education and community engagement are highly underrepresented aspects in evaluating higher education sustainability programs. Yarime and Tanake [14] conclude that we lack concepts and methodologies for evaluating higher education sustainability programs on how they design their education and community engagement. Little is known about the learning objectives, educational activities and assessment methods appropriate in transdisciplinary programs, which makes implementing them, and assessing their effectiveness, a challenge [11]. As such, Remington-Doucette and colleagues [7] posed the question: What do students learn in transdisciplinary education, and what is the quality of their learning?

This contribution aims to grasp what students actually learn when working in transdisciplinary projects. The theoretical framework adopted is that of boundary crossing, in which boundaries between practices, that is, different disciplines, perspectives, cultures or societal groups, are viewed as the most powerful places for learning, knowledge co-creation and innovation [15,16]. 'Boundaries' refer to socio-ecological and socio-cultural differences between diverse knowledge or professional domains, leading to discontinuity in action or interaction [17] (p. 1011). Learning requires students to cross the boundaries and build bridges between the practices around the boundary. Boundary crossing theory describes this learning by four learning mechanisms, i.e., identification, coordination, reflection and transformation. These learning mechanisms—in short—relate to identifying your own expertise and perspective, and the expertise and perspectives of other stakeholders involved in the problem (i.e., identification), organising and facilitating effective collaboration between people across boundaries (i.e., coordination), perspective-making between different practices and learning from each other's perspectives on the problem at hand (i.e., reflection), and co-creating new knowledge by combining the different perspectives around the boundary to develop a more sustainable practice (i.e., transformation). These are four learning processes that, when adopted effectively, are leverages for learning across boundaries. They apply to all transdisciplinary problems, independent of the specific content or pursued solution for the sustainability problem at hand, and as such, are generically applicable in transdisciplinary education.

This study asks the question, "How can the boundary crossing framework, and its underlying four learning mechanisms, be made instrumental to explicate, stimulate and value what students (can) learn in transdisciplinary sustainability projects?" To answer this question, this design-based study focuses on student learning and working in twenty transdisciplinary sustainability projects from different life science higher education institutions. The study used multiple methods and multiple rounds of data collection and interventions [18,19] to iteratively develop and refine the Boundary Crossing Rubric. The Boundary Crossing Rubric (further BC-rubric) operationalises the four learning

mechanisms in criteria and performance levels, describing concrete student behaviour representative of student–stakeholder collaboration and learning in transdisciplinary projects on sustainability issues. The instrument aims to be subject insensitive (or generalisable), meaning that it is a generic instrument that can be used in all kinds of transdisciplinary projects in which students learn and co-create with multiple stakeholders, independent of the specific content or context of the problem. This instrument was reviewed and refined in three meetings with teachers from different educational levels, resulting in a final version of the BC-rubric. Additionally, these teachers' meetings also discussed and identified the potential use of this instrument for designing, facilitating and assessing transdisciplinary learning for sustainability.

The next sections first describe the challenge of making transdisciplinary learning for sustainability visible. The boundary crossing framework, including its four learning mechanisms, is then posed as a framework that can help to make visible the student–stakeholder collaboration processes that are at the heart of transdisciplinary learning. An argument is then made for why we developed a rubric as an instrument to visualise boundary crossing learning, and how this instrument is expected to be stimulating for sustainable learning. The method section first describes the context of this study, being the 'regional learning environment' (RLE) [18,19]. The RLE is an exemplary transdisciplinary learning environment, in which student groups, often from different disciplines and/or cultures, learn and work with stakeholders from outside the university on real world regional sustainability problems, with the ultimate aim of contributing to sustainable development of that region. In the RLE, the Boundary Crossing Rubric is developed in an iterative design process of mixed method monitoring of twelve, and then intervening in eight, RLEs.

*1.1. Making Transdisciplinary Learning Visible*

Remington-Doucette et al. [7] posed the question of what students actually learn in transdisciplinary courses. This is a challenging and intriguing question, both in designing learning environments for this learning and in assessing this learning [5]. Transdisciplinary sustainability problems and the sustainability competencies that people need to deal with these problems [8] are vague and indefinite. Thus, the 'what' of learning, that is, "What is it that we want our students to learn, and when are they successful in doing this?" is difficult to define. Scardamalia et al. [20] argue this is inherent to what they call 'knowledge building environments' (p. 234), which are needed in sustainability education or education for 21st century skills. These environments challenge students to adopt activities and learning processes, leading to learning outcomes that often remain obscured in learning environments and assessments used in higher education institutions. We call these learning outcomes 'learning surprises' [21,22]. Sustainability education should allow for these 'learning surprises', or as Scardamalia and colleagues [20] say, "breakthroughs in education for 21st century skills require integrating two different approaches: working backwards from goals and the emergence of new competencies" (p. 231). Thus, the learning goals and success criteria for transdisciplinary learning are vague and partly open. To avoid ambiguity and in search for standardisation or controllability, educational programs often stick to well-structured, simple, tame problems that do have clear-cut, right answers [5]. Cremers [23] was one of the first to design and study hybrid, transdisciplinary learning environments, in which students and regional partners learn and work together on authentic problems embedded in the region. These learning environments explicitly aim to stimulate innovation and knowledge creation. As these environments offer ample opportunity for learning surprises and individual leaning trajectories, Cremers explicitly chose not to define predefined learning outcomes. Instead, she asked students to justify what they learned and how they developed themselves while working in the transdisciplinary setting. This turned out to be very difficult both for students and teachers. They could not give words to what was learned, and they were both inclined to focus mainly on learning disciplinary content knowledge instead of reporting on learning outcomes resulting from learning and working with different stakeholders. This example shows the challenge of making transdisciplinary learning visible. A comparable problem is found

in broad entrepreneurship education. Broad entrepreneurship is about identifying, evaluating and exploiting opportunities to create economic, societal, cultural and/or ecological value for somebody else. This is also a transdisciplinary process contrived with uncertainty and complexity, in which the outcomes or values are not always known or even measurable [24]. Dealing with these uncertainties and the chaotic and risky process typifies that of the most effective entrepreneurs [24]. From entrepreneurship education, we can learn that assessments should not focus on ticking off clear-cut and standardised levels of competencies, but on providing insight into the processes students undertake in the entrepreneurial process of creating value for others [22,25]. Lönngren and Svanstöm [5] also argue that innovative assessment for valuing student transdisciplinary learning in sustainability education is lacking hitherto, and Kalsoom [26] and Bramwell-Lalor [27] discuss the challenges in assessing sustainability competencies and using assessments to stimulate sustainable learning. Yarime and Tanaka [14] explicitly stressed that assessment tools in sustainability education should be sensitive to the collaborative processes students undertake with other stakeholders.

Initiatives for studying the design and assessment of transdisciplinary sustainability education review written work and mainly address cognitive processes. In the last decade, various assessment rubrics have been developed for learning outcomes related to collaborative or interdisciplinary settings, such as sustainable engineering [28,29], interdisciplinary learning [30] and intercultural learning [31]. All these instances review student written work, and as such focus on cognitive processes, thereby missing the complexity of collaborating and co-creating new, innovative knowledge across disciplines and stakeholders. Lönngren et al. [28] aimed to operationalise the sustainability competencies of Wiek and colleagues [8] into learning objectives and an assessment rubric to assess student learning while working on sustainability problems. Again, the actual collaboration processes were left out of their research, as they also reviewed higher education student written responses to wicked sustainability problems. Remington-Doucette and colleagues [7] also aimed to assess student performance in three of these sustainability competencies (system thinking and normative and strategic competence) by reviewing student responses to case studies. However, while some case studies involve going into the real world, many case studies provide a limited form of interaction between students and stakeholders outside the educational institution. Trencher et al. [32] assessed the extent to which sustainability programs fostered the development of sustainability competencies via student and teacher questionnaires. Results revealed that most programs were research-oriented and mainly fostered theoretical and conceptual knowledge, while practice-oriented programs demonstrated higher success in building interpersonal, strategic and normative competencies. Student reports highlighted the demand for more practice-oriented education, with specific focus on more transdisciplinary projects for and with real societal stakeholders. Yarime and Tanake [14] reviewed sixteen assessment tools in sustainability programs, and also concluded these are largely insensitive to student-stakeholder collaborative processes, while they also stressed that sustainability education increasingly involves transdisciplinary cooperation between students and societal partners. In addition, Remington-Doucette and colleagues [7] make a strong plea for student–stakeholder collaboration in real-life transdisciplinary sustainability problems, and they argue that this is what makes transdisciplinary courses different from interdisciplinary education. Innovative methods for making transdisciplinary learning and competence for sustainability tangible are required to help sustainability education and research move a step further. This study tries to bridge this gap by operationalising the student–stakeholder collaboration activities that typify transdisciplinary learning and working. To do this, the boundary crossing theory is used as theoretical framework. This will be discussed in the next section.

*1.2. Theoretical Framework: Boundary Crossing*

Wicked sustainability problems cannot be solved alone. Actually, they cannot be solved at all, and because they have no definite answer, they can be dealt with in a more optimal or transformational manner [1]. This requires a transdisciplinary approach, in which stakeholders from different academic disciplines and from the wider community engage in knowledge co-creation, change and

transformation [33]. For this to happen, stakeholders have to cross each other's boundaries. A 'stakeholder' at this place is defined as a person or party with an interest or concern in an issue at hand [34]. 'Boundaries', understood as the explication of socio-cultural and socio-ecological differences between people and their practices, are often perceived and experienced as difficult and hampering learning and action [15,16]. The greatest barriers, as perceived by participants in inter- and transdisciplinary projects, are the additional time needed, coping with different traditions and a lack of common terminology. Furthermore, many hurdles arise from agreeing on a common problem formulation and the lack of personal chemistry [35]. The boundary crossing theory stresses that boundaries are the most powerful places for learning, but that crossing them does not come easily and requires explicit support [15,36].To make use of the transformative potential of boundaries, and overcome its barriers, people should develop 'boundary crossing competence', i.e., the ability to work and communicate across different practices and become transformation agents [37].

Education for sustainability should foster student ability to identify boundaries, explicitly approach them, cross them and try to span them to create new, transformative practices across boundaries and with stakeholders of diverse perspectives around these boundaries (e.g., [38]). Innovative assessments in sustainability education should explicitly value and stimulate these student–stakeholder processes [2,14]. In their review on boundary crossing, Akkerman and Bakker [15] identified four learning mechanisms that should take place at the boundaries to exploit their potential. When used appropriately, these four learning mechanisms can be used as leverage for learning and working across boundaries required for co-creating new knowledge and practices for sustainable development. The first learning mechanism is called 'identification.' This involves the questioning of the own and others' core identities, and the mutual complementarity of different practices. Identification leads to insights into what the diverse practices concern, not necessarily into actual collaboration. 'Coordination', the second mechanism, expresses what people learn from seeking communicative connections between diverse practices or perspectives, e.g., by contacting each other to exchange relevant information, or by using languages from different practices. These connections can be established by effective means and procedures, also called 'boundary objects' [15] (p. 133), that allow different practices to communicate efficiently in distributed work. Where coordination takes place, dialogue between parties is established only as far as necessary to maintain the work flow. 'Reflection', the third mechanism, contains perspective-making and -taking. People come "to realize and explicate differences between practices and thus to learn something new about their own and others' practices" [15] (p. 144). 'Transformation', the fourth learning mechanism, involves joint work at the boundaries between practices, combining ingredients from different practices into something new (i.e., hybridisation). Transformation results in new knowledge creation, innovation and, ideally, changes to existing practices or to new, hybrid, more sustainable practices. Figure 1 [15,39] visualises these four learning mechanisms and operationalises them in several questions that trigger these learning processes.

The theoretical concept of boundary crossing and its four learning mechanisms are argued to be a good lens for understanding learning that occurs when people learn across practices (e.g., [23,40,41]). Akkerman and Bakker subdivided the learning mechanisms further into various sub-processes (see [15]) (p. 151). Initial attempts have been made to operationalise the learning mechanisms, using the sub-processes, in coding schemes for research [23,42]. In this study, we aim to operationalise the four learning mechanics in terms of student behaviour when learning and working on sustainability problems in multi-stakeholder, transdisciplinary settings. We argue these learning processes are representative of this learning independent of context, content or sustainability issue at hand. As this operationalisation results from real student behaviour in real transdisciplinary projects in sustainability education [7,32], we argue that the Boundary Crossing Rubric will be instrumental for designing, stimulating or coaching, along with assessing student sustainable learning independent of content [20].

| Visualisation of the learning mechanism | Aim of the learning mechanism | What questions to ask yourself to stimulate the learning mechanism |
|---|---|---|
| | **Identification** Gaining insight into complementarity and added value of the different practices around the boundary | • What expertise do I have? • What expertise do I lack in the context of the sustainability problem at hand? • Who are the stakeholders? • What is their expertise, stake and perspective? • How to they relate to each other? |
| | **Coordination** Collaboration to deal with the problem, but geared towards efficiency and working along each other (e.g. task division) | • How can I involve the different stakeholders? • How do I approach the different stakeholders? • How can we communicate and collaborate effectively? • What agreements do we make with each other? • What object can I use or develop to facilitate mutual communication |
| | **Reflection** Learning to see the problem through the eyes of another. Both defining and exchanging perspectives focused on mutual meaning making and connecting different perspectives and expertise. | • How do I help other stakeholders understand my perspective? • What can I learn from the perspectives of the other stakeholders involved? • What can we learn from each other? |
| | **Transformation** Development of new knowledge/practices; an end result that could not have been developed without actual collaboration and integration of perspectives. | • What is my vision on the new practice? • How can we combine our knowledge and perspectives into a (innovative, but realistic) solution? • How can I get others enthusiastic for this new practice? • How can I stimulate follow-up to build on the new practice (towards a sustainable new practice)? |

**Figure 1.** Visualisation of the boundary crossing learning mechanisms.

An additional argument for developing a rubric is that, when designed properly [43] a rubric offers ample opportunities for the following: (1) making transparent what has to be/can be learned; (2) explicating student development; (3) providing proper feedback; (4) conducting self- or peer-assessments; (5) setting personal learning goals; and (6) coaching student development. These

processes should be part of sustainability education, making students resilient for their unknown future [27,44].

Aimed at developing a rubric in a design-based iterative process, the following research questions guided this study:

1.  How can the boundary crossing theory and its underlying learning mechanisms be operationalised in observable student behaviour in relation to student–stakeholder interaction in such a way that it helps design, stimulate and assess student learning with and from 'the other' in transdisciplinary sustainability education?
2.  How do teachers perceive the developed instrument and its value for designing, facilitating and assessing student sustainable learning?

## 2. Materials and Methods

This section first describes the regional learning environment (RLE). This exemplary transdisciplinary learning environment for sustainable development is the context of this study. Second, the phases of this design-based study are described: a first round of monitoring twelve RLEs; a second round of intervening and monitoring in eight RLEs; and finally, the iterative design of the actual Boundary Crossing Rubric. In this iterative design, several teachers' workshops were conducted for the design, validation and refinement of the rubric (research question 1), as well as for identifying its perceived value for design, facilitation and assessment of student sustainable learning (research question 2).

### 2.1. Context of the Study: The Regional Learning Environment

Regional learning environments were set up in the Dutch life sciences sector by educational institutes in collaboration with various regional stakeholders [19,45,46]. These parties collaboratively developed a long-term regional knowledge agenda to foster sustainable development of the region. Many Dutch institutions for higher and vocational education incorporated RLE projects, originating in this knowledge agenda, into their curricula. Examples of study programs involved are Environmental Sciences, Land and Water Management and Climate Studies. The general aim of the RLE is twofold, namely (1) to support students' and other parties' learning through working on wicked, authentic sustainability problems, and (2) to contribute to sustainable regional development. From an educational perspective, the RLE is an authentic, demand-driven collaborative and multi-stakeholder learning environment (e.g., [19,47]) that inherently requires students to adopt boundary crossing learning processes. In the RLE, students work in groups on real world regional, i.e., supra-local, problems with various regional stakeholders like local and regional authorities, semi-governmental bodies, entrepreneurs, research institutes, NGOs and citizens [45,46]. All stakeholders have an interest in the problem at hand. Solving the problem requires integration and/or co-creation of new knowledge between students and multiple regional stakeholders. Teachers in RLEs facilitate and/or coach the learning process, rather than transferring knowledge as an expert. Additionally, the teacher is a 'learner' himself, working in an almost equal relationship with the students to collaboratively tackle complex regional problems. The end result is meant to be of value for the external commissioner and to contribute to sustainable regional development or even transformation. As such, the RLE inherently confronts various boundaries to be crossed. First, there are disciplinary boundaries, as usually students work in multidisciplinary groups and have to work with experts from various disciplines, including teachers. Second, there is the university–society boundary, as students always work on a problem commissioned by one of the regional societal partners. Additionally, students mostly have to collaborate and co-create with other societal stakeholders with a stake in the problem at hand. Third, cultural boundaries also play a role in some RLEs when students from different cultural backgrounds have to work together—for example, international students who have to work with Dutch societal partners, or sometimes students who go abroad during their RLE project. Observing

student learning and working in the RLE offers ample opportunity for operationalising what boundary crossing activities in sustainability education look like.

*2.2. Design of the Study: Using The RLE to Grasp Boundary Crossing Learning*

To answer our research questions, we went through a design-based, iterative process. In a first attempt to grasp what students learn when working on a sustainability problem in a multi-stakeholders context, twelve RLEs in Dutch life science education were monitored in the first round. As this first round showed that much boundary crossing activity was far from optimally stimulated in the monitored RLEs, the second round developed an intervention to explicitly stimulate students to adopt boundary crossing behaviour in their collaboration with (multiple) stakeholders. Monitoring student learning and behaviour in an additional eight RLEs supported by the intervention allowed for a better operationalisation of the boundary crossing behaviour required for developing the 'Boundary Crossing Rubric' (BC-rubric). Two teacher meetings reviewing and discussing the developed instrument resulted in a final version of the rubric, as well as indications on its use for designing, coaching and assessing student learning in transdisciplinary sustainability education. This method section describes the iterative development of the Boundary Crossing Rubric, resulting in a final version of the rubric in the result section.

In the first round, twelve RLEs implemented in different Dutch life science education programs were monitored: five in academic study programs (n = 233) and seven in professional higher education programs (n = 135). In the second round, an additional eight RLEs (N = 122) from five Dutch life science education programs participated in the intervention study: one in academic education (n = 12) and seven in professional higher education programs (n = 110). Teachers of the RLEs were contacted and voluntarily agreed to participate, allowing us to monitor their RLEs. All RLEs met the educational characteristics as described above, and as such represented transdisciplinary learning environments requiring students to collaborate and co-create across boundaries for sustainable regional development. Students in all RLEs worked in student groups of mostly five or six students. In fourteen RLEs, these were multidisciplinary student groups, that is, students from different study programs working together, while in six RLEs, students of the same study program worked together. Each student group worked on a different project assignment, commissioned by a societal party, of which the results contributed to the sustainable development of the respective region. For a more elaborate description of the participating RLEs see [18,19].

2.2.1. First Round. Lack of Boundary Crossing Activity

The first round showed that, even though all RLEs were transdisciplinary learning environments offering ample opportunities for multi-stakeholder co-creation of new knowledge and more sustainable practices for the region, this multi-stakeholder learning was not at all optimally stimulated and valued [18,19]. Course documents showed that only in one RLE, learning with and from other practices was explicitly addressed, stimulated and assessed. In this one instance, this learning was explicated in learning objectives, and was addressed and reflected upon in coaching sessions with the student groups and evaluated in a final assessment rubric. In all the other RLEs, the learning objectives and assessment criteria only reflected the use or development of disciplinary knowledge. Actual interaction, collaboration, let alone co-creation, with external stakeholders was often not observed in these RLEs. Students tried to find all required information on the stakeholders and their perspectives through the internet, and, in some cases, teachers allowed students to contact external partners only through them. However, teachers univocally reported learning with and across practices and co-creating new sustainable ideas to be one of the main reasons for working with RLEs.

To grasp teachers' ideas about what students should be able to learn from working in the multi-stakeholder RLE, we conducted a semi-structured workshop with 25 teachers participating in different RLEs, including eight of the twelve RLEs monitored in this first round. This workshop focused on explicating what students do and can learn in the RLE that results from its boundary

crossing transdisciplinary nature. Building on teachers' experiences and reflecting on them using the four boundary crossing learning mechanisms, we collaboratively developed success criteria for 'being a good boundary crosser in multi-stakeholder learning settings' [48]. To guide this discussion, we posed the question "What performances differentiate a student who is successful in crossing boundaries between external stakeholders from less successful students?" [49]. These descriptions were used to describe the 'highest performance level' of the to-be developed Boundary Crossing Rubric.

The data collected in the first round did not allow for a proper operationalisation of student boundary crossing activities in transdisciplinary multi-stakeholder settings. The gap between 'the highest level' of boundary crossing performance identified in the teacher workshop and actual student behaviour seen in the RLEs was too wide to bridge, simply because students were not challenged to adopt boundary crossing behaviour in their RLEs. Therefore, in a second round of this design study, an intervention was developed to explicitly trigger students to adopt behaviour related to the identification, coordination, reflection and transformation–learning processes of boundary crossing.

### 2.2.2. Second Round. Boundary Crossing Intervention: Student–Stakeholder Workshops

To trigger students to actively adopt boundary crossing learning mechanisms in their working with others, two four-hour student–stakeholder workshops were developed [42]. These workshops, facilitated by both researchers, were plugged into student RLEs, and by extension were an integral part of their learning and working in the RLEs. The first workshop contained several specific assignments triggering the identification and coordination processes. Examples of activities were stakeholder analyses explicating information about each stakeholders' expertise, perspective and stake, or preparing a stakeholder event. This workshop was integrated into the beginning phase of student projects. The second workshop, integrated into a later stage of the RLE, specifically focused on the reflection and transformation processes. Examples of activities were a reflective review of identified and experienced stakeholder perspectives so far, and a brainstorm on an impactful, stakeholder collaborative activity meant to trigger transformation in the region (see [42] for an elaborate description of the workshops and the intervention study). During the further RLE project process, actual RLE teachers kept on referring to the outcomes of the workshops when guiding student RLE learning processes.

To grasp representative student behaviour for the four learning mechanisms, students were observed as they undertook the workshop activities. Additionally, students answered three open-ended questions at the end of the RLE project. These questions asked students to describe (1) their undertaken activities in learning and working with multiple stakeholders, (2) their experiences in terms of peaks and troughs in this learning and (3) their intentions for how to approach a new, comparable transdisciplinary multi-stakeholder project.

The observations and student answers to the above questions were coded in a deductive multi-rater coding process. A coding scheme was developed based on the boundary crossing learning mechanisms (see [42]) and the success criteria for 'a good boundary crosser' identified in the first round. This allowed identification of various concrete examples of student behaviour in these four learning mechanisms and their underlying sub-processes. These descriptions gave ample opportunities for both performance indicators and distinctive levels of performance for a Boundary Crossing Rubric.

### 2.2.3. The Iterative Design of the Rubric

The Boundary Crossing Rubric was designed by the researchers in a set of collaborative working sessions, discussing all data and using the boundary crossing framework of Akkerman and Bakker [15] and its four learning mechanisms, as an analytical framework. A rubric describes several performance indicators (in the left column) combined with quality definitions for those indicators at different performance levels (i.e., the other columns) [43]. The first round of this design study resulted in the collaborative development of a set of qualifications describing 'the successful boundary crosser.' This list of qualifications was used to build a first version of the rubric, as this list describes the highest

performance levels expected from students at various performance indicators. These performances were linked to the boundary crossing learning mechanisms to specify the performance indicators that should be present in the rubric (i.e., the left column). These indicators were used to grasp concrete boundary crossing activities that students undertook in rounds 1 and 2. The variety of data collected allowed the researchers to explicate three 'lower' level performances per performance indicator, representing different growth levels of student performances in multi-stakeholder learning for sustainable (regional) development.

Several groups of teachers were involved in the iterative development, validation and refinement of the BC-rubric (research question 1), and for identifying its perceived value (research question 2). The first version of the Boundary Crossing Rubric was reviewed in a workshop with twenty teachers experienced with RLEs in either vocational, higher professional or academic education. In this meeting, facilitated by two researchers (i.e., the two authors), three activities were undertaken. First, participants reviewed the readability and understandability of the instrument. Second, teachers were asked to recall a specific RLE project, and filled in the rubric for the following questions: what do you see your students do during their multi-stakeholder projects, and what would you like to see your students do during their projects? Third, teachers were asked what utilities of the BC-rubric they saw for their own education. The findings of the first teacher workshop were implemented in a second version of the BC-rubric, reviewed in a second workshop with eight teachers. Six teachers worked in higher professional education, two in vocational education from seven institutions. These teachers all had experience with transdisciplinary projects, being the RLE or a comparable learning environment (e.g., [23]; Hybrid learning arrangement). This mixed group of teachers made it possible to explore the usability of the BC-rubric for transdisciplinary projects other than the RLE. In this workshop, these activities were undertaken: first, teachers explicated the main dilemmas they experienced when working in the transdisciplinary learning environments. This activity was undertaken to afterwards evaluate if teachers felt the BC-rubric offered possible solutions to these dilemmas, showing the usefulness of the rubric. Second, the rubric was clarified and discussed in an interactive presentation. Third, teachers filled in a questionnaire evaluating seven statements on the usefulness of the rubric on a five-point Likert scale (1 = disagree; 5 = agree). These statements evaluated the extent to which the rubric was perceived as useful for the following: giving words to learn with (and from) people of other practices; setting learning goals; formative assessment; summative assessment; and development of learning lines through the curriculum, coaching and communicating with external parties. This provided information on the extent to which teachers felt the rubric was useful for designing, coaching or assessing boundary crossing learning. Fourth, in response to the evaluation questionnaire, teachers were asked if they saw additional applications of the rubric. This activity also referred to their identified dilemmas to see if the rubric could be helpful for dealing with these dilemmas, and/or if the rubric could be adjusted slightly to do so.

A final refinement and validation step was the involvement of two university teachers teaching in an utmost transdisciplinary master course (the European Workshop') [50] as part of an Environmental Sciences programme at a life science university. In the European Workshop, students worked in multidisciplinary groups on a transdisciplinary sustainability problem commissioned by an international societal stakeholder. Teachers were struggling for a long time with how to better stimulate student learning and working in these contexts with the many present boundaries. These teachers became enthusiastic about the rubric and wanted to test it in their master course. In collaboration with these two expert teachers, two types of refinement were carried out: (1) vocabulary changes to better fit student vocabulary and (2) references to 'external stakeholders' were replaced by 'others/other people/other practices', to make the rubric applicable to different boundaries. The current rubric focused on students learning and working with external stakeholders. However, boundary crossing can also refer to learning and working across disciplinary, cultural and national boundaries, as in the master course 'European Workshop.' As such, this final step helped to make the BC-rubric more

generically applicable to learning situations in which students have to cross different boundaries while working on a sustainability challenge.

## 3. Results

For the first research question, this section first describes the collaboratively developed success criteria for 'being a good boundary crosser' in transdisciplinary sustainability contexts and their relation to the boundary crossing learning mechanisms. Then, the final Boundary Crossing Rubric (BC-rubric) will be presented. To answer the second research question, the qualitative and quantitative findings from teacher evaluations are presented.

### 3.1. Success Criteria for 'the Good Boundary Crosser'

In the first teacher workshop, teachers were challenged to explicate the behaviour they would want their students to show in their RLE project related to learning and working with multiple stakeholders. The boundary crossing learning processes were used to guide this explication. Table 1 shows the resultant list of success criteria for being a good boundary crosser in transdisciplinary settings like the RLE. This table also shows how these criteria link to the theoretical boundary crossing learning mechanisms and how they are translated into criteria for the final BC-rubric (Table 2). Stimulating teachers to explicate concrete student behaviour in the RLE allowed operationalisation of the theoretical learning mechanisms into more specific performance indicators as the basis for the criteria of the rubric (i.e., the left column of Table 2).

**Table 1.** Success criteria describing performances of 'the successful boundary crosser'.

| A Student Who Is a Good 'Boundary Crosser' . . . . | Link to Boundary Crossing Learning Practice [15] | Translation into Criterion for the BC-Rubric |
|---|---|---|
| shows that (s)he is interested in the project not only to pass the course (a good grade), but also to deliver an end result that can be applied in practice and is useful for other people; | Transformation | Intention to develop a sustainable solution (transformation 1) |
| considers what expertise is needed to execute the project successfully and what the limitations and contributions are of his/her own expertise; | Identification | Knowing your own practice (identification 1) Other practices needed (identification 2) |
| is open to learning from and contacts other people, sees the advantage of using other people's expertise; | Reflection | Reconsider perspective (reflection 1) Learn from other (reflection 2) |
| facilitates and stimulates the collaboration of people involved in the project; | Collaboration & reflection | Collaborates with different people (coordination 2) Stimulates others to learn (reflection 3) |
| empathises with other people's perspectives/interests/ideas, also when they differ from his/her own; | Reflection | Learn from others (reflection 2) |
| actively searches for ways to learn from others, and encourages other people to reflect and to learn as well; | Reflection: | Reconsider perspective (reflection 1) Stimulate others to learning (reflection 3) |
| explicates how multiple perspectives, interests and expertise are used and integrated in the project to deliver a better end result; | Transformation | Integrate different practices into a new practice (transformation 3) |
| explicates how the end result can be implemented in practice, and which steps to be taken to do so; | Transformation | Envisioning the new sustainable practice (transformation 2) Thinking of a follow-up |
| shows enthusiasm and effort to be actively involved in follow-up activities. | Transformation | Thinking of a follow-up (transformation 4) |

**Table 2.** Boundary crossing rubric: a tool to support inter- and transdisciplinary learning in sustainability education.

| | D<br>The Student... | C<br>The Student... | B<br>The Student... | A<br>The Student... |
|---|---|---|---|---|
| **Identification 1:**<br>Identify one's own expertise and one's own limitations | does not explicate which expertise (s)he possesses and which expertise might be missing to execute the project successfully | explicates his/her own expertise in terms of knowledge, skills and network that can contribute to the project | previous cell<br>+<br>identifies his/her own limitations regarding expertise needed to execute the project. | relates his/her own expertise to that of the other members of the project team and maps what kind of expertise is missing to execute the project successfully |
| **Identification 2:**<br>Identify other perspectives relevant for the project and problem at hand | does not actively explore other perspectives | shows awareness of various perspectives, but does not explicitly address these different perspectives in the light of the project | identifies people, including their interests, perspectives, expertise and mutual relations, relevant for executing the project | previous cell<br>+<br>the student explicates for which aspects of the project he/she needs other people and plans actions to contact these other people |
| **Coordination 1:**<br>Contact other people | takes no action to contact other people<br>or<br>takes action, but only because it is a requirement of the course | contacts a few other people close to the problem and easy to address (e.g., given by the teachers).<br>prefers to contact external people in a digital way | develops active and face to face contact with relevant other people | initiates and organises collaborative meetings with relevant other people with the intention to collaboratively share ideas, develop new ideas and tune own ideas |
| **Coordination 2:**<br>Collaborate purposefully with other people | does not actively and purposefully collaborate with other people<br>or<br>is merely frustrated by the challenges that emerge in this collaboration | carries out activities to discuss a few other perspectives, closely related to his/her own background | aims at purposeful collaborations with various relevant people to the project. Discovers and/or contributes to developing a boundary object (BO) relevant for people involved to facilitate collaboration for executing the project | previous cell<br>+<br>uses the BO actively to accommodate multi-, inter- or transdisciplinary collaboration, and checks whether everybody contributes to the project. If not, (s)he takes action |
| **Perspective-making and learning from each other 1:**<br>(Re)consider perspectives | considers the project purely from his/her own perspective and interest | shows limited openness to other perspectives relevant for the project and/or, considers the input from other perspectives mainly for his/her own benefit (i.e., what can I use from you?) | actively explicates and/or discusses various perspectives relevant for the project and searches for ways to combine perspectives (i.e., how can the different perspectives contribute to and strengthen the project) | previous cell<br>+<br>explicates how other perspectives influenced his/her own perspective on the project |
| **Perspective-making and learning from each other 2:**<br>Learn from other people | merely aims to complete the project, not to learn from other people (i.e., shows no learning attitude at all) | reflects on own learning process and development specifically and can explicate these | explicitly shows the willingness to learn from other people during the project | actively searches for ways to learn from others and purposefully develop him/herself |
| **Perspective-making and learning from each other 3:**<br>Stimulate others to learn (general) | shows no action in stimulating other people to learn from each other | reflects with team members on each other's role, contribution and development during the project, but does not actively transfer the results into improved performance of other people during the projects | initiates reflective actions between people involved in the project aimed at learning from the project (both process and content-wise) | previous cell<br>+<br>actively encourages other people's learning in light of the project |

**Table 2.** *Cont.*

| | D<br>The Student... | C<br>The Student... | B<br>The Student... | A<br>The Student... |
|---|---|---|---|---|
| **Transformation 1:**<br>(start)<br>Intend to develop a new, sustainable practice | shows an attitude of conducting the project solely to pass the course | shows an attitude to want to develop a project result that serves a limited amount of perspectives | shows an attitude to want to develop a project result that serves multiple perspectives | previous cell<br>+<br>shows an attitude of wanting to deliver a project result that is innovative or inspiring innovation |
| **Transformation 2:**<br>(process)<br>Envision new practices during project process | has difficulty and/or shows no interest to think out-of-the-box. Sticks to mainly traditional or obvious solutions | tries to include innovative elements in traditional solutions | shows out-of-the-box thinking, serving multiple perspectives through weighing pros and cons of various possible solutions | previous cell<br>+<br>clarifies a vision for the new to be developed practice, i.e., can explicate how the new practice would look, how it functions and what to be done to realise this new practice |
| **Transformation 3:**<br>(product)<br>Integrate various perspectives, interests or expertise in a final product | shows merely a compilation of insights of students involved in the final project. Does not explicate the integration of multiple perspectives, interests or expertise | shows how own ideas and those of other students are integrated in the final product.<br>Shows insights in how other perspectives are integrated and how realistic the final product is in practice | shows convincingly how (s)he weighted multiple perspectives and interests in the final product, and considers its practical and its innovative character | previous cell<br>+<br>clearly explicates how to effectively inform other external people involved about the outcome of the final product |
| **Transformation 4:**<br>(follow-up)<br>Stimulate a follow-up on project results | finishes the project for school and shows no interests in follow-up activities | finishes the project and mentions a few options for follow-up activities | finishes the project, explicates how it can be implemented in practice and which steps to be taken to do so | previous cell<br>+<br>shows enthusiasm and effort to be actively involved in follow-up activities |

### 3.2. Translation into the Final Rubric

The review and revision rounds resulted in a final BC-rubric thought to be generically applicable to students learning and working across different boundaries on sustainability issues. The boundary crossing theory and underlying learning mechanisms turned out to be a very helpful theory for explicating this rubric. It goes too far to address all vocabulary changes and adaptions carried through in the rounds, except for one. The final discussion with the two expert teachers from the master course 'European Workshop' led to renaming the learning mechanism 'reflection' [15] to 'perspective-making and learning from the other.' The argument for this change was that 'reflection' constantly led to another association and interpretation different to that intended in the boundary crossing theory. Reflection in education is mostly interpreted as a personal and individual activity of looking back on your own experiences and actions and learning from them for the next time [51]. However, reflection in the boundary crossing theory refers to rethinking one's own ideas and perspectives in the light of the perspectives and ideas of other people around the boundary. It is also about stimulating others to undertake this perspective-making and taking process. The rewording of the learning mechanism 'reflection' should make its meaning more boundary-crossing-specific.

Table 2 shows the final BC-rubric. This rubric shows in the left column various performance indicators based on the four BC learning mechanisms operationalised into several performances identified (see Table 1). Every performance indicator is described on four growth levels, representing different levels of performance observed in the monitored RLEs.

### 3.3. The Usability of the BC-Rubric for Designing, Facilitating and Assessing Sustainability Learning

To answer the second research question on the usability of the BC-rubric, the teachers' inputs were reviewed. In the first workshop, we asked teachers to use the rubric to mark for all performance indicators (1) what do you see your students do during their multi-stakeholders projects (actual behaviour)? and (2) what would you like to see your students do during their projects (desired behaviour)? This activity showed big differences between these two scorings. This supported our observations of the twelve RLEs in the first round of this design study, in which learning across boundaries was not explicitly stimulated in the RLE projects. Learning across the boundaries was expected to happen automatically, but this turned out to not be the case. On the other hand, all teachers expressed this boundary crossing learning to be the ultimate added value for student learning in these transdisciplinary settings. So, the main conclusion of this first teacher workshop was that the BC-rubric allowed for making explicit what all teachers implicitly felt to be the added value and learning potential of learning in the RLEs. The instrument allowed for (1) describing specific learning objectives geared to learning across boundaries and (2) coaching students while working in their RLE project and explicitly reflecting with students on what and how they learn from all the people involved.

In the second teacher workshop reviewing the rubric, some quantitative and qualitative data were collected concerning teachers perception of the usability of the rubric for designing, facilitating and assessing boundary crossing learning. Table 3 shows that teachers were positive about the utilities of the rubric, and Table 4 shows some teacher quotes supporting or explaining these positive evaluations.

**Table 3.** Teachers' rating of the usability of the BC-rubric (Likert scale 1–5).

| Question<br>To What Extent Is the BC-Rubric Usable for . . . | Means *<br>(N = 8) | Remarks |
| --- | --- | --- |
| Giving words to learning with and from others | 4.4 | Helps to develop a shared vision with colleagues<br>Giving words to what students learn |
| Formative assessment | 4.6 | Makes assessment and reflection much more clear and transparent |
| Summative assessment | 4.1 | Not before this process is embedded in our education first<br>When design, learning tasks and assessment are aligned<br>Makes explicit for what aspects students should provide specific evidence/justification (e.g., in their portfolios)<br>It allows for assessing the process students go through during working with other practices |
| Coaching | 4.4 | Gives handles for coaching discussions |
| Formulating learning outcomes | 4.3 | Aspects of the rubric can be linked to our qualification profile<br>Allows for setting personal learning goals within a shared framework |
| Communicating with external stakeholders | 4.3 | Helpful for expectation management |
| Developing a BC learning trajectory | 3.7 | Boundary crossing is only taught near the end of educational programs |

* items were scored on a Likert scale from 1 'totally disagree' to 5 'totally agree.'

**Table 4.** Teacher quotes on the usability of the BC-rubric.

| |
| --- |
| "This [rubric] finally allows me to give words to what I see my students do and learn while working in our hybrid learning environment." (Teacher A, professional higher education) |
| "Fantastic this is a generic instrument. These aspects of learning and working together can be applied to all our authentic projects to a more or lesser degree. That makes this instrument usable for all these project, and still allows for individual learning paths." (Teacher B, professional higher education) |
| "I would also like to use this as a design instrument. If you want to stimulate students to adopt these processes, you need to design learning activities in such a way that they stimulate students to actively use them." (Teacher A, professional higher education) |
| "This instrument offers ample opportunities for aligning design and assessment." (Teacher C, professional higher education/vocational education) |
| "These aspects can easily be linked to our end qualifications and give theses concrete meaning. For example the qualifications 'networking' or 'innovative capacity.'" (Teacher D, professional higher education) |
| "You can use this instrument to let students consciously reflect on learning and working together with external stakeholders, for example vial self or peer-assessments." (Teachers E, vocational education) |
| "This allows for making self-assessments and reflections more concrete. I would ask my student to assess themselves using the rubric and justify their scoring with concrete examples of performed activities." (Teacher D, professional higher education) |
| "This instrument allows for discussing student attitudes of 'I do this because I have to or because it is graded.' We want our students to develop an attitude showing 'I want to develop an innovative idea for the region in collaboration with the external stakeholders.'" (Docent D, professional higher education) |
| "I think this is also very usable in vocational education. Our students are often better networkers than we are. This instrument allows for explicitly stimulating and valuing competencies that belong to this." (Teacher F, vocational education) |
| "I would use this instrument as a starting point for (re)designing our education." (Teacher G, professional higher education) |

## 4. Discussion

This design-based study aimed to develop an instrument to grasp what students can learn in sustainability education in which they work on sustainability challenges with multiple (external) stakeholders. Sustainability challenges are wicked problems without a definite solution, and in which conflicts of interests among multiple stakeholders are inevitable [1]. Dealing with these problems requires people to cross the boundaries between the different practices that represent different perspectives on this problem. Sustainability education needs to educate students to deal with these wicked problems and the boundary crossing processes involved. This requires what Scardamalia and colleagues [20] call 'knowledge building environments' (p. 234), which are geared towards knowledge co-creation, innovation and transformation, and that are partly and inevitably

unpredictable. Even though sustainability education, or transdisciplinary education, is emerging in higher education [3,11], it still often 'prepares' students for simple, well-structured problems. This is likely because of the challenge of dealing with unpredictability [5,7], and because teachers are also not educated in a transdisciplinary manner [4]. Additionally, many sustainability programs are missing real-life collaboration with societal partners [14,32,52]. Grasping what and how students learn in transdisciplinary learning environments is a challenge [7]. This design-based study aimed to develop a generic instrument, building on the boundary crossing theory and its four learning mechanisms [15], to make transdisciplinary learning visible and to facilitate designing, coaching and assessing sustainable learning. This resulted in the Boundary Crossing Rubric (BC-rubric, Table 2).

The BC-rubric tries to grasp and operationalise "What students can learn when learning, working and co-creating new knowledge together with different practices." The BC-rubric is developed in the regional learning environment (RLE) as an exemplary transdisciplinary learning environment in life science education. The RLE requires students to cross boundaries between disciplines and academia/society to develop innovative practices for sustainable regional development. Using document analyses, teacher workshops, workshop activities supporting students to adopt boundary crossing activities, observing students and analysing their learning reports, we iteratively developed the BC-rubric. Characteristic of and unique to the BC-rubric is, first, that, contrary to most other rubrics (e.g., [28,30]), it describes actual and observable student behaviour in interaction with others in sustainability education [7,14]. This inevitably requires sustainability education, both in design and assessment, to incorporate actual collaboration between students and other, external, stakeholders [2,32]. Second, by taking the boundary crossing theory and its four learning mechanisms (i.e., identification, coordination, reflection and transformation) as a framework, and reviewing and validating the BC-rubric with teachers working in non-RLE transdisciplinary university master courses [50], we argue that the BC-rubric is a generic instrument. The BC rubric is applicable to all kinds of authentic multi-stakeholder learning environments in which students learn, work and co-create with other practices to deal with sustainability challenges, irrespective the content of the challenge or the specific disciplinary learning outcomes of the program. Third, this rubric describes learning outcomes in terms of processes instead of predefined and standardised cognitive or competency outcomes or levels that all students need to fulfil. While these processes apply to all transdisciplinary problems in which students have to cross boundaries, how and to what extent students adopt these processes can differ, depending, for example, on the complexity of the problem, the number of stakeholders involved, or the differences experienced between the stakeholders. This makes the rubric suitable for 'knowledge building environments' [20] that are geared towards knowledge co-creation, innovation and transformation in which learning outcomes are often unpredictable.

Teachers experienced with the RLE or comparable learning environments were enthusiastic about the instrument for its use in their own educational practice (see Tables 3 and 4). The most reported response was "this gives words to what I see my students do that neither they nor I could previously put into words." Teachers see many opportunities for using the BC-rubric in the formative assessment process [39]: setting (personal) learning goals, giving words to development, coaching further development, feedback and self/peer assessment. As such, the BC-rubric allows for individual learning paths and fostering sustainable learning for individual students [27,44]. Teachers also perceive the BC-rubric as an educational design instrument, and, as such, as an instrument that allows development of sustainability programs in which learning goals, learning activities and assessments are aligned [53], and making sustainable development becomes a more explicit aspect of higher education curricula [9].Viewing the rubric as an educational design instrument might offer opportunities for building concepts on what education and community outreach in higher education sustainability programs looks like [13,14]. Teachers in our study reported seeing linkages between the performance indicators of the rubric and the end qualifications of their programs. This last evaluation is important for the feasibility of the BC-rubric to be adopted by education programs. This is an important step towards more and more explicitly embedding multi-stakeholder learning in transdisciplinary

or sustainability education [2,14] and paving the way for stimulating students to develop boundary crossing competence [37]. This sustainability education is expected to better prepare students for dealing with an unpredictable future, filled with sustainability challenges that require innovative solutions for which many boundaries are to be crossed.

*First Experiences with Implementing the BC-Rubric in Education for Sustainability Development*

The BC-rubric was tried out in the university master course European Workshop [50]. In this eight week, full-time course, the BC-rubric was used for the following: (1) to let students conduct a starting self-assessment on the rubric performance indicators and set personal learning goals; (2) to coach student groups while learning and working together in their multidisciplinary groups and with various external stakeholders; and (3) to let students reflect on their boundary crossing learning after completing the project. The first experience showed that it was difficult for students to understand the instrument and set specific learning goals regarding 'working with the other.' On the other hand, their final reflections on what they learned from working with other disciplines, cultures and societal partners seemed more specific and diverse than reflections in previous cohorts. Teachers reported that the BC-rubric allowed them to coach student groups much more on their group process. Teachers were specifically enthusiastic about the fact that the rubric allowed them to make students more aware of their own boundary crossing behaviour. While some students were reluctant to cross the boundaries, and showed behaviour hampering the boundary crossing process, other students were natural 'brokers' [15], playing a critical and connecting role in the multidisciplinary and cultural student groups as well as with external stakeholders. The BC-rubric allowed teachers to both have a more critical discussion with the more reluctant students and to explicitly value the critical and connecting behaviour of the brokers and make these students aware of their connecting role. This is a nice example of 'it gives words to what I see happening.' On the other hand, teachers also needed to gain more experience in application and understanding of the BC-rubric and its underlying theory to more optimally use it in their coaching of student groups. Teachers are also not used to looking at student performances this way, or coaching them in learning from others around the boundary. This supports one of the important recommendations of Filho et al. [4] for the future of sustainability education in higher education, namely that teacher education should train professionals in an interdisciplinary manner. The Boundary Crossing Rubric might also be a useful instrument for this purpose. Teachers were also struggling with the 'transformation' performance indicators (see Table 2; Transformation 1,2,3,4). They doubt whether the transformation BC process as described in the BC-rubric is possible or feasible in education, even in this academic master course. This is an interesting avenue for future research and practice in sustainability education. We are running a research project that studies the implementation, use and effects of the BC-rubric more in-depth. We intend to identify how to best support the implementation and use of the instrument to most optimally stimulate student sustainable learning. This, in turn, can allow teachers to coach and value the diversity of student behaviour in sustainability challenges in which students learn with a variety of (external) stakeholders towards a more sustainable solution.

**Author Contributions:** Both authors worked in close collaboration and were equally involved in the conceptualisation, design and data collection of the study as well as in the data analysis resulting into the iterative design of the BC rubric. J.G. took the lead in writing the manuscript, while C.O. thoroughly reviewed, edited and refined the manuscript leading to the final manuscript.

**Funding:** This research received no external funding.

**Conflicts of Interest:** The authors declare no conflict of interest.

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
