# Peer review of "Towards a Rubric for Stimulating and Evaluating Sustainable Learning"

_sustainability, doi:10.3390/su11040969_

Round 1

Reviewer 1 Report

A concise abstract which is not always easy but the use of the word 'wicked'didn't feel quite right for me and is it academia versus society or mofre of vested interests and fake news against difficult to come to terms with science?  One to ponder!

Lines 52 - 56 sign post well.

Para starting line 57 - can I suggest this is too long and could be broken at line 76.

line 61 - good to have such a helpful and concise definition.

line 63 - felt it would be worth mentioning Scardamalia at this point although good to see her later referenced in line 112.

line 141 - another very concise and helpful sentence.

end of line 158 - I found this sentence starting with Again to be a bit awkward though - any chance of improving?

line 172 - projects I think rather than project

line 213 - their own I think

Figure 1 was very helpful and badly needed by this time - thank you!!

Around the 290 mark I wondered if you might think about adding gthe idea of social pedagogy and Storo and the 'common third' which could be of interest here?

line 405 felt a bit dense - can you unpack a little?

Good to see Table 1 at this point which begged the quesiotn of a Table 2 which was very helpful indeed.

line 480 - renaming of 'reflection' was excellent I thought and helpful.

line 501 - sentence starting at the end of this line needs looking at again I think.

line 502 - leave the word 'learning' out so that it starts the para after the table.

Table 4 needs re-formatting I suggest.

lines 605 - 611 were very helpful and convincing I felt.

line 614 - I thought the transformation rubric was excellent and somwhat surprised that teachers had diffuclty with this.  I wonder if what might be missing here was some sense of urgency and related social agency (Hayward, B, 2012) in relation to the joint projects to really find solutions to pressing problems.  In other words to give the transformation some real bite.  Certainly I would support further research in this area.

Thank you for an excellent paper.

Author Response

A concise abstract which is not always easy but the use of the word 'wicked'didn't feel quite right for me and is it academia versus society or mofre of vested interests and fake news against difficult to come to terms with science?  One to ponder!

Answer: Both reviewers were intrigued by the work “wicked”. We feel it denotes several typical features of sustainability problems. We incorporated more explanation of what we mean by wicked, referring to the work of Arjen Wals who is demarcating wicked problems from simple and complex problems (Wals 2014;2015). The features of wicked problems inherently lead to conflicting ideas of various stakeholders and the necessity to effectively collaborate and cross boundaries between the perspectives of these stakeholders to co-create a possible way of dealing with the issue.

The explanation we added immediately in the second sentence of our revised manuscript is as follows:

We call these problems “wicked” problems (Wals, 2015) stressing that both the problem as well as its solution(s) are not clear and keep changing whenever we try to define them, like global warming, food security or biodiversity loss. Wals (2015, p 17) demarcates wicked problems from simple and complex problems. According to Wals, wicked problems are characterized by their resistance to define them, having no right or wrong answers, and their unfamiliar, ambiguous, chaotic nature in which conflicts of interests among multiple stakeholders is inevitable (Ceulemans, Molderez, and Van Liedekerke 2015). Dealing with these wicked issues requires collaboration and meaning making between these different stakeholders (Wals, 2014).  (line 30-37).

To avoid misunderstandings, we deleted the word wicked out of the abstract, as it is not crucial to mention it here at that place.

See also our response to reviewer 2 on the “wicked” issue.  

Lines 52 - 56 sign post well. Thank you

Para starting line 57 - can I suggest this is too long and could be broken at line 76. Done as suggested

line 61 - good to have such a helpful and concise definition. Thank you

line 63 - felt it would be worth mentioning Scardamalia at this point although good to see her later referenced in line 112. answer: we did not really see the direct link to the work of Scardamalia in line 63. Therefore we chose to leave it as it was.

line 141 - another very concise and helpful sentence. Thank you

end of line 158 - I found this sentence starting with Again to be a bit awkward though - any chance of improving? Answer: We rephrased the sentence as “Again, the actual collaboration processes are left out of their research, as they also reviewed higher education students’ written responses to wicked sustainability problems.”. This can be found in line 178-180 of the revised manuscript

line 172 - projects I think rather than project. Adjusted as suggested

Figure 1 was very helpful and badly needed by this time - thank you!! Thank you too

Around the 290 mark I wondered if you might think about adding gthe idea of social pedagogy and Storo and the 'common third' which could be of interest here? Answer: We see the link / similarities between the idea of boundary crossing/learning together and building a boundary object (a common third) that brings people around the boundary (in social pedagogy the client/child and the professional) together and fosters building their relationship and their learning together. However, we decided not to make use of this concept in our text as we feel that many readers of SE might not be familiar with social pedagogy, common third and child-professional activities. And as such, incorporating this viewpoint might confuse readers instead of make things more clear. Additionally, this special issue focussed on higher education, while the common third is often related to child/youngsters and professional collaborations, which might add to the confusion. We hope that you can agree with our consideration and decision to not include this in our text.

line 405 felt a bit dense - can you unpack a little? Answer: We elaborated this part of the text a bit to make it hopefully more clear. Though, you might need Table 2 to really understand this part of the text. (if the reader might not be so familiar with rubrics or with how you can design a rubric)

Good to see Table 1 at this point which begged the quesiotn of a Table 2 which was very helpful indeed. Thank you

line 480 - renaming of 'reflection' was excellent I thought and helpful. Glad to hear this!

line 501 - sentence starting at the end of this line needs looking at again I think. We reshuffled and rephrased this sentence to make it more clear: “This supported our observations of the twelve RLEs in the first round of this design study, in which we saw that learning across boundaries was not explicitly stimulated in the RLE projects. Learning across the boundaries was expected to happen automatically, but this turned out to not be the case. While, on the other hand, all teachers expressed this boundary crossing learning to be the ultimate added value for students learning in these kind of transdisciplinary settings.” (line 522-527 in the revised manuscript)

line 502 - leave the word 'learning' out so that it starts the para after the table. Done.

Table 4 needs re-formatting I suggest. Answer: We used the formatting prescribed in the template of this journal. We are happy to change this lay-out but for now know not how to approach this.

lines 605 - 611 were very helpful and convincing I felt. Thank you

line 614 - I thought the transformation rubric was excellent and somwhat surprised that teachers had diffuclty with this.  I wonder if what might be missing here was some sense of urgency and related social agency (Hayward, B, 2012) in relation to the joint projects to really find solutions to pressing problems.  In other words to give the transformation some real bite.  Certainly I would support further research in this area.  Answer: we linked this finding to previous research (mentioned in the call of the special issue, Filho, Manolas and Pace (2015)) also addressing that one of the barriers for bringing this kind of learning into higher education is that teachers themselves are not educated in this (inter/transdisciplinary manner). Indeed an intriguing area for further study.

Thank you for an excellent paper. Thank you for this great compliment

Reviewer 2 Report

I really appreciated this work because it attempts to utilize research methods and understandings of research language into a format that demonstrates the complexity of writing about a subject in a way that does not "necessarily fit" the standard format. I struggled somewhat with the article just as I struggle with my own work in attempting to "break the mold" while still aiming to get a piece published. While I believe I understood the use of the word "wicked," I would like to see either an explanation of the word in context of where it came from and/or in the vocabulary/scenario in which it is defined or described by the youth participants and/or the researcher(s). I personally would have appreciated hearing/reading more of the voices of the students to demonstrate how and what this process helped them learn (but perhaps that is another article/publication). I personally have a love/hate relationship with rubrics as they seem  in many instances to stifile creativity and thought; however, I do appreciate the effort and ideas that were suggested to make this particular rubric flexible yet structural.

Author Response

Review

I really appreciated this work because it attempts to utilize research methods and understandings of research language into a format that demonstrates the complexity of writing about a subject in a way that does not "necessarily fit" the standard format. I struggled somewhat with the article just as I struggle with my own work in attempting to "break the mold" while still aiming to get a piece published. While I believe I understood the use of the word "wicked," I would like to see either an explanation of the word in context of where it came from and/or in the vocabulary/scenario in which it is defined or described by the youth participants and/or the researcher(s). I personally would have appreciated hearing/reading more of the voices of the students to demonstrate how and what this process helped them learn (but perhaps that is another article/publication). I personally have a love/hate relationship with rubrics as they seem  in many instances to stifile creativity and thought; however, I do appreciate the effort and ideas that were suggested to make this particular rubric flexible yet structural.

Answers:

Both researchers were struggling with the word “wicked”.  We explained what mean by this term by framing it in the work of Professor Wals who is an influential scholar in the field of learning for sustainability. He positions sustainability problems on a simple-complex-wicked dimension. We added this explanation in the beginning of the manuscript (line 30) to immediately frame the kind of sustainability problems we are talking about in this paper. We hope this is insightful for the reviewers and readers of SI.

With respect to the student voice, we can imagine that reviewer 2 would like to hear more student voice in our research. We certainly do too. But this is indeed a follow up article that we are doing research for as we speak.

We also agree and feel for the love/hate struggle of the reviewer with rubrics. While they provide transparency, they are also found to hamper creativity and might even stimulate “teaching and learning for the rubric”. This obviously is not a practice that we would like to stimulate. For this reason we see most opportunities for the rubric to help explicate, discuss and further stimulate student learning. This as a formative assessment instrument instead of a summative instrument demarking what all student should be able to do a t a certain point.

We really appreciated the reviewers observation of the struggle to break the mold and get your work published. We are very happy that we managed to find a proper way to find a balance between these two. Thank you very much for your support in this respect.